# Characterization of *C9orf72* haplotypes to evaluate the effects of normal and pathological variations on its expression and splicing

**Israel Ben-Dor**[1], **Crystal Pacut**[2], **Yuval Nevo**[3], **Eva L. Feldman**[2], **Benjamin E. Reubinoff**[1]*

**1** The Hadassah Human Embryonic Stem Cell Research Center, The Goldyne Savad Institute of Gene Therapy, Hadassah Medical Center, Jerusalem, Israel, **2** Department of Neurology, University of Michigan Medical School, Ann Arbor, MI, United States of America, **3** Computation Center, Hebrew University–Hadassah Medical School, Jerusalem, Israel

* benr@hadassah.org.il

**Data Availability Statement:** The RNA-seq data has been deposited at the GEO database, accession number GSE162943 (http://www.ncbi.nlm.nih.gov/geo).

## Abstract

Expansion of the hexanucleotide repeat (HR) in the first intron of the *C9orf72* gene is the most common genetic cause of amyotrophic lateral sclerosis (ALS) and frontotemporal dementia (FTD) in Caucasians. All *C9orf72*-ALS/FTD patients share a common risk (R) haplotype. To study *C9orf72* expression and splicing from the mutant R allele compared to the complementary normal allele in ALS/FTD patients, we initially created a detailed molecular map of the single nucleotide polymorphism (SNP) signature and the HR length of the various *C9orf72* haplotypes in Caucasians. We leveraged this map to determine the allelic origin of transcripts per patient, and decipher the effects of pathological and normal HR lengths on *C9orf72* expression and splicing. In *C9orf72* ALS patients' cells, the HR expanded allele, compared to non-R allele, was associated with decreased levels of a downstream initiated transcript variant and increased levels of transcripts initiated upstream of the HR. HR expanded R alleles correlated with high levels of unspliced intron 1 and activation of cryptic donor splice sites along intron 1. Retention of intron 1 was associated with sequential intron 2 retention. The SNP signature of *C9orf72* haplotypes described here enables allele-specific analysis of transcriptional products and may pave the way to allele-specific therapeutic strategies.

## Author summary

Amyotrophic lateral sclerosis (ALS) and frontotemporal dementia (FTD) are progressive neurodegenerative diseases, whose most frequent genetic cause is hexanucleotide repeat (HR) expansion from normal 2 to 20 repeats to pathological hundreds of repeats within a non-coding region of the *C9orf72* gene. Haplotype is a specific combination of multiple polymorphic sites along a chromosome that are inherited together in block. We characterized the single nucleotide polymorphism (SNP) signature and HR length of the major

**Funding:** This work was supported by the following donations: A donation from the late Alfred Taubman (B.R. and E.L.F). The Sinai Medical Staff Foundation (E.L.F) A donation from Judy and Sidney Swartz. The Swartz Foundation (B.R.) Legacy Heritage Fund LTD. (B.R.) The funders had no role in the study design, data collection and analysis, decision to publish, or preparation of the manuscript.

**Competing interests:** The Authors have declared that no competing interests exist.

*C9orf72* haplotypes in Caucasians to identify the allelic origin of *C9orf72* transcripts per patient and determine the effects of expanded HR on *C9orf72* gene expression and splicing. In *C9orf72* ALS patients' cells, the HR expanded allele, compared to non-R allele, was associated with decreased levels of downstream initiated transcript variant, increased levels of upstream initiated transcripts, accumulation of introns 1 and 2, and abnormal splicing at cryptic splice sites along intron 1. The *C9orf72* haplotypes DNA signatures described here are valuable for studying C9-ALS/FTD pathogenesis and for developing allele-specific therapeutic strategies.

## Introduction

Amyotrophic lateral sclerosis (ALS [OMIM: 612069]) and frontotemporal dementia (FTD [OMIM: 600274]) are two fatal neurodegenerative diseases. ALS is characterized by motor neuron (MN) degeneration in the brain and spinal cord, which causes progressive muscle wasting and paralysis. FTD is characterized by neuronal degeneration in the frontal and temporal brain lobes, leading to deterioration in language, behavior control, and emotional management. About 10% of ALS and 30–50% of FTD cases have a family history of the disease, typically inherited in an autosomal dominant fashion [1,2].

GGGGCC hexanucleotide repeat (HR) expansion in chromosome 9 open reading frame 72 (*C9orf72*) gene is the most common genetic cause of ALS and FTD in Caucasians [3,4]. It accounts for around 39% of familial and 7% of sporadic ALS cases and 25% of familial and 6% of sporadic FTD cases in the Caucasians [5]. The HR is located between two alternative 5' non-coding exons, 1a and 1b (S1 Fig). In approximately half of all alleles, the HR is repeated twice, and in over 98% of the alleles its length is less than 17 repeats [3,6,7]. In diseased alleles, HR length is typically hundreds or thousands of repeats, while intermediate lengths of repeats (disease threshold of ~30) are relatively rare [3,4,8]. In the nucleus, HR expansion is bidirectionally transcribed to sense and antisense transcripts that sequester crucial RNA binding proteins [9]. In the cytoplasm, these transcripts undergo repeat-associated non-ATG (RAN) translation, resulting in the production and accumulation of pathogenic dipeptide repeat proteins [10]. *C9orf72* expression from the mutant gene is variable and depends on its methylation status [11–14]. Hypermethylation occurs in 36% and 17% of C9orf72-ALS and FTD patients, respectively [15], but not in normal and intermediate repeat lengths [14,16].

Toxic mRNA and dipeptide buildup concurrent with reduced levels of normal C9orf72 protein [17,18] interferes with basic cellular functions, such as splicing, translation, nucleocytoplasmic transport, autophagy, and organelle structure and function, eventually leading to neurodegeneration [19,20].

The *C9orf72* gene contains two alternative promoters, three alternative first exons, and two alternative last exons in the three annotated transcript variants: V1 (NM_145005), V2 (NM_018325), and V3 (NM_001256054). V1 and V3 share a common start site, and their first exon ends at two alternative splice sites upstream to the HR. Hence, after V1 and V3 are transcribed, the HR is removed by post-transcriptional splicing. V2, the major *C9orf72* variant that normally accounts for 85–95% of *C9orf72* transcripts [13,21], starts downstream of the HR, which is therefore not present in transcribed V2 (S1 Fig). The majority of *C9orf72* transcripts carry 11 exons that encode a 481-amino acid protein, while a fraction of the transcripts carry 5 exons with an alternative polyA site at an extended exon 5, which encodes a 222-amino acid protein (S1 Fig).

It has been demonstrated that all *C9orf72* ALS and FTD (C9-ALS/FTD) patients share a common haplotype that is also abundant in healthy individuals. This risk (R) haplotype was initially characterized by a single nucleotide polymorphic site (SNP) rs3849942G>A, located 3.2 kb downstream to the *C9orf72* gene 3' terminus [3,22]. The risk haplotype was further characterized by 20 SNPs in a 120 kb block that spans *C9orf72* with two additional genes [23,24], while more detailed maps demonstrated 44 and 82 SNP signatures along 110 kb [25,26]. Altogether, these maps include 19 SNPs within the 27.3 kb transcribed region of *C9orf72* (underlined in Table 1), some are R haplotype specific SNPs.

Here, we established a map of over 100 SNPs and indels along the transcribed region of *C9orf72* and identified the SNP signature of the R haplotype, as well as the other major haplotypes in Caucasians. This enables identification of the allelic origin of *C9orf72* transcripts in patient and control cells. We show that pathological expansion of the HR is associated with lower V2 levels, while the V1 and V3 levels are comparable with normal R alleles. We also show that pathological HR expansion lowers *C9orf72* first- and second-intron splicing efficiency and increases aberrant splicing at first-intron cryptic donor splice sites. We also determined the repeat length that characterizes each haplotype and show that increased HR length in the normal range is associated with higher *C9orf72* expression. Deciphering the genetic profile of *C9orf72* haplotypes and characterizing deleterious transcripts from the mutant gene are invaluable for developing disease allele-specific DNA or RNA targeting therapeutics.

## Results

### Characterization of *C9orf72* haplotypes

We sought to create a detailed molecular map for the *C9orf72* haplotypes to enable biallelic expression analysis and study the effects of HR length on its expression and splicing. Since most of the C9-ALS patients are Caucasians, we focused our efforts on Caucasian Americans. We first analyzed 12 C9-ALS patients for the surrogate marker rs3849942-A (S1 Table, #26). Eleven of these C9-ALS patients were heterozygous (A/G), while one patient (D-184) was homozygous (A/A). This patient carried an R haplotype with a pathological HR expansion (hereafter called $R_d$; d = disease) and an R haplotype with a normal HR length (hereafter called $R_h$; h = healthy). By directly sequencing PCR products along the *C9orf72* locus in this patient (S1 Table, #1–26), we determined the SNP composition of the R haplotype, while the full SNP signature was accomplished by deep sequencing data (Tables 1 and S2).

We further confirmed the R haplotype and characterized the complementary non-R haplotype in the 11 heterozygous (A/G) C9-ALS patients by the same methodology. Six of the patients (D-007, D-540, D-836, D-338, D-431, D-478) shared a common haplotype that we named "K", carrying the previously described SNP rs10757668 G to A substitution in the 5' untranslated region (UTR) of exon 2 [3]. Four of the patients (D-312, D-805, D-733, D-850) shared a common haplotype that we named "F" and one patient (D-883) carried a rare haplotype that we named "N". Similar analysis of 42 control subjects confirmed these haplotypes and revealed three additional haplotypes that we named P, J, and Q (S3 Table). The full characterization of these haplotypes was accomplished by deep sequencing data (Tables 1 and S2).

The distribution of the normal haplotypes among our 54 Caucasian American subjects was 28.4% for $R_h$, 22.1% for F, 20% for K, 16.8% for P, and 7.4%, 3.2%, and 2.1% for the haplotypes J, N, and Q, respectively (Fig 1A, left chart). Only one allele was identified with a "hybrid" haplotype, in which F was replaced by K towards the 3' end of the gene (H-157 in S3 Table). Each haplotype, except Q, had three or more haplotype-specific SNPs (marked in italics in Tables 1 and S2). According to 1000 Genomes data, the frequency of these SNPs in Europeans is similar along the entire *C9orf72* locus and particularly between exons 1 and 7 (Fig 1B). Based on these

**Table 1. Detailed genetic map of the major *C9orf72* haplotypes.**

| SNP ID rs# | Haplotype | | | | | | | | SNP ID rs# | Haplotype | | | | | | | |
|---|---|---|---|---|---|---|---|---|---|---|---|---|---|---|---|---|---|
| | R | F | K | P | N | J | Q | Z | | R | F | K | P | N | J | Q | Z |
| 41272893 | A | A | A | A | *G* | A | A | A | 3849944 | T | T | C | C | C | C | C | C |
| 78074330 | A | A | A | A | *G* | A | A | A | 10967985 | G | G | *A* | G | G | G | G | G |
| HR | | 2 | 2 | 5 | 4 | ~6 | 2 | | 774358 | A | *T* | A | A | A | A | A | A |
| 117462033 | C | C | C | T | C | C | T | C | 28526385 | T | T | T | C | C | C | C | C |
| 112048460 | C | C | C | C | *T* | C | C | C | 774357 | *A* | G | G | G | G | G | G | G |
| 2282240 | C | *T* | C | C | C | C | C | C | 1565948 | G | G | A | A | A | A | A | A |
| 2282241 | C | A | C | A | C | C | A | A | 774356 | C | T | T | T | C | T | T | T |
| 4520261 | C | C | C | C | *T* | C | C | C | 7860526 | A | A | *G* | A | A | A | A | A |
| 17696653 | T | T | T | T | *C* | T | T | T | 2297694 | C | C | C | G | G | G | G | G |
| 3849946 | C | A | C | A | C | C | A | A | 10812614 | A | *T* | A | A | A | A | A | A |
| 145319692 | *6* | 5 | 5 | 5 | 5 | 5 | 5 | 5 | 4879566 | T | T | T | C | C | C | C | C |
| 78900326 | C | C | C | C | *T* | C | C | C | 4879565 | T | T | T | A | T | A | A | A |
| 700824 | T | T | C | T | T | C | T | T | 67245195 | A | A | A | A | A | *T* | A | A |
| 700825 | G | G | G | G | G | *A* | G | G | 10757665 | T | T | *C* | T | T | T | T | T |
| 4879572 | A | A/G | A | A | A | A | A | A | 4879564 | C | C | C | T | C | T | C | T |
| 76602706 | T | T | T | T | *C* | T | T | T | 62538126 | C | C | C/A | C | C | C | C | C |
| 13284967 | C | T | C | T | C | C | T | T | 112616482 | A | A | A | A | *C* | A | A | A |
| 111630075 | G | G | G | G | *A* | G | G | G | 700828 | C | T | T | T | C | T | T | T |
| 10812619 | T | C | T | C | T | T | C | C | 10812613 | T | T | *C* | T | T | T | T | T |
| 72710405 | G | G | G | *A* | G | G | G | G | 17835861 | C | C | C | G | G | G | C | G |
| 111653040 | A | A/G | A | A | A | A | A | A | 34670748 | G | G/A | G | G | G | G | G | G |
| 3849945 | *G* | A | A | A | A | A | A | A | 111253152 | T | T | T | T | *A* | T | T | T |
| 72710403 | A | A | A | *G* | A | A | A | A | 1022902 | G | G | G | A | G | A | G | A |
| 77534147 | AC | AC | AC | AC | AC | – | AC | AC | 17769246 | T | T | T | T | *G* | T | T | T |
| 7875392 | C | C | *G* | C | C | C | C | C | 188460764 | T/A | T | T | T | A | T | T | T |
| 17769300 | G | G | G | G | *C* | G | G | G | 76412392 | T | T | T | T | T | T | T | *A* |
| 7872223 | C | C | *A* | C | C | C | C | C | 10967984 | C | T | C | T | C | T | T | T |
| 10967988 | G | T | G | T | G | G | T | T | 10967983 | T | T | *A* | T | T | T | T | T |
| 10757669 | G | G | *A* | G | G | G | G | G | 60613335 | A | A | A | A | *G* | A | A | A |
| 41272891 | G | G | G | G | *A* | G | G | G | 35815580 | G | G | G | G | G | *A* | G | G |
| 10757668 | C | C | *T* | C | C | C | C | C | 12686452 | T | T | T | C | T | C | C | C |
| 2120721 | G | C | G | C | C | C | C | C | 10124158 | T | T | G | G | G | G | G | G |
| 1031153 | C | *T* | C | C | C | C | C | C | 12349820 | T | T | *C* | T | T | T | T | T |
| 10757667 | A | A | *C* | A | A | A | A | A | 80272464 | G | G | *C* | G | G | G | G | G |
| 10967986 | G | A | G | A | G | A | A | A | 113076260 | T | T | T | T | *C* | T | T | T |
| 10757666 | A | A | T | A | T | A | A | A | 68005046 | G | G | G | G | G | *A* | G | G |
| 10812618 | T | T | *G* | T | T | T | T | T | 4878487 | T | T | T | C | T | C | T | C |
| 142843265 | – | AG | AG | AG | AG | AG | AG | AG | 71510499 | C | C | C | C | C | *T* | C | C |
| 2492816 | A | G | G | A | A | A | G | A | 10967981 | T | C | T | T | T | T | C | T |
| 7859060 | A | A | *G* | A | A | A | A | A | 12347201 | T | T | *C* | T | T | T | T | T |
| 7874565 | T | T | *C* | T | T | T | T | T | 117867610 | C | C | C | C | C | C | C | *T* |
| 10812617 | T | T | *C* | T | T | T | T | T | 2453565 | *T* | C | C | C | C | C | C | C |
| 7858531 | A | A | *G* | A | A | A | A | A | 62538125 | A | A | *C* | A | A | A | A | A |
| 2453555 | *A* | G | G | G | G | G | G | G | 10120735 | T | T | T | T | *C* | T | T | T |
| 2484319 | *C* | A | A | A | A | A | A | A | 773723 | G | G | T | T | G | T | T | T |
| 10441712 | T | T | C | C | C | C | T | C | 10967979 | A | A | *C* | A | A | A | A | A |

*(Continued)*

**Table 1.** (Continued)

| SNP ID rs# | Haplotype | | | | | | | | SNP ID rs# | Haplotype | | | | | | | |
|---|---|---|---|---|---|---|---|---|---|---|---|---|---|---|---|---|---|
| | R | F | K | P | N | J | Q | Z | | R | F | K | P | N | J | Q | Z |
| 10812616 | T | T | A | A | A | A | T | A | 72727512 | C | C | C | G | C | C | G | C |
| 10812615 | T | T | C | C | C | C | T | C | 2305045 | A | A | A | A | A | A | A | *C* |
| 34366576 | A | A | A | G | G | G | A | G | 73440933 | A | A | A | A | *G* | A | A | A |
| <u>2453554</u> | *T* | C | C | C | C | C | C | C | 200583482 | - | - | - | - | *A* | - | - | - |
| 17769294 | T | T | T | *C* | T | T | T | T | 3739526 | G | G | G | T | G | T | G | T |
| <u>774359</u> | C | T | T | T | C | T | T | T | <u>13691</u> | G | *A* | G | G | G | G | G | G |
| 12347222 | G | G | G | T | G | T | T | T | <u>9103</u> | A | A | *G* | A | A | A | A | A |

The map includes SNPs at the transcribed region of *C9orf72* with a minor allele having a frequency of 1% or more within the European population. The map also includes the Z haplotype SNPs. Haplotype-specific SNPs are indicated in italic bold font, and sub-haplotype variations are signified by two alternative nucleotides in the same table cell (*e.g.*, rs4879572 in haplotype F). SNP ID corresponds to the dbSNP database (www.ncbi.nlm.nih.gov/SNP). SNPs that were described in previous SNP profiling of the R haplotype are highlighted by underline. The *C9orf72* gene is located at the (-) strand and therefore the bases of *C9orf72* transcripts are complementary to the bases in the table. SNPs position in the *C9orf72* gene and on chromosome 9, their flanking sequence, and their frequency in the global population and in Europeans are provided in S2 Table, which also includes the 5' and 3' flanking regions of the gene.

data, we predicted haplotype frequency among Europeans (Fig 1A, right chart), which was comparable to the haplotype distribution in our sample of Caucasian Americans (Fig 1A, left chart). Sequencing analysis of two PCR-amplified products in intron 1 (503 bp) and intron 6 (666 bp), which together carry nine polymorphic sites (S1 Table, #9, 21), suffices to distinguish between these seven haplotypes (Table 1). Notably, we identified an additional haplotype that we named "Z" from available deep sequencing data. We verified its sequence in several Asian samples from the 1000 Genomes Project, which we identified as homozygous for this haplotype [*e.g.*, NA21124 (Gujarati) and HG00473 (Han Chinese)]. The Z haplotype contains three unique polymorphic sites: rs76412392T>A, rs117867610C>T and rs2305045A>C (Tables 1 and S2), and is rare in Europeans (~0.5%), and thus was not represented in our sample. Distinguishing between Q and Z haplotypes requires PCR analysis in addition to the two described above.

Altogether, we created a detailed molecular map for eight haplotypes at the *C9orf72* locus that enables identification of the allelic origin of transcripts that carry heterozygous SNP sites.

The HR length may play a role in *C9orf72* expression. We therefore completed our molecular map by analyzing the HR length. Previous studies have shown that HR length in normal R haplotype ($R_h$) is usually longer than non-R haplotypes (rs3849942 A/A versus G/G) [3,6,7,25]. Yet, the specific SNP signature and identity of the various non-R haplotypes was not known in these studies, so their HR length could not be determined. We therefore analyzed the HR length in Caucasian American subjects for whom we had identified their haplotype, as described above (Fig 1A, left chart), and in additional cell lines by PCR using primers flanking the HR [3]. Matching between the haplotype and HR length was initially determined in the 12 C9-ALS patients carrying a single normal allele copy (S2B and S2C Fig). As expected, we easily determined the length of the normal allele in each patient, while the expanded allele was detected in only one patient (D-478) as sawtooth peaks of 20–35 repeats with a maximal peak at 27 repeats (S2C Fig). For simplicity, we hereafter refer to this length as "intermediate", and to larger alleles that were undetectable by PCR as "large expansions". We next determined the HR length in 15 Caucasian subjects with homozygous haplotype (S2A Fig) and/or with equal HR length in both alleles, which were represented by a single peak. In another 30 subjects in whom HR length could not be absolutely linked to a specific haplotype, the linkage between

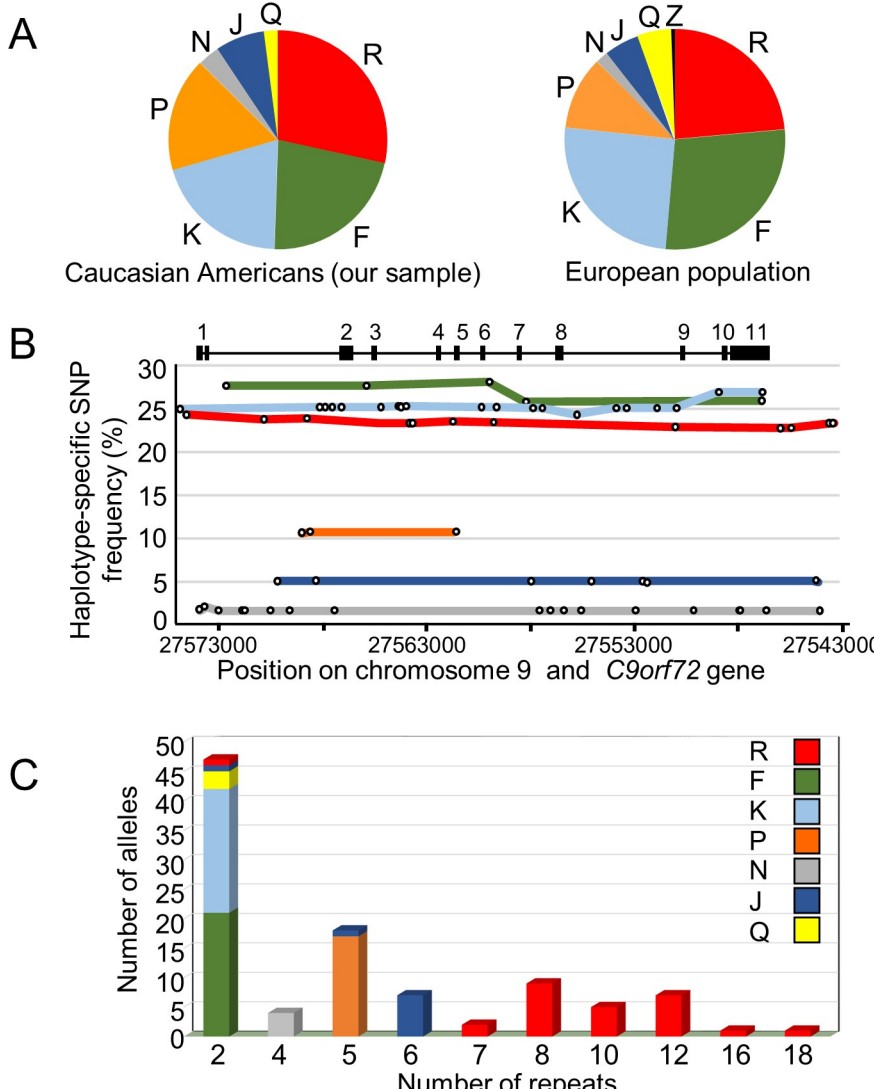

**Fig 1. *C9orf72* haplotype frequency and HR lengths. (A)** *C9orf72* haplotype frequency. The left chart represents the distribution in our sample of 54 Caucasian Americans (95 normal non-mutated alleles) described in S3 Table. The right chart represents the distribution in Europeans, based on haplotype-specific SNP frequency according to the 1000 Genomes data (viewed in Ensembl, http://www.ensembl.org/Homo_sapiens/Search/Results?q=; facet_feature_type=; site=ensembl;facet_species=Human;page=1). The Q haplotype does not harbor haplotype-specific SNPs; therefore, its frequency was calculated from the frequency of SNPs that are shared by Q and an additional haplotype with a known frequency. **(B)** Frequency of haplotype-specific SNPs along the transcribed region of *C9orf72* gene and its 5' and 3' flanking regions among Europeans (S2 Table). SNPs are represented by dots, and their position in the graph indicates their position on chromosome 9 and in the *C9orf72* gene (*x*-axis), and their frequency within the European population (*y*-axis). The SNPs of each haplotype are connected by a line with the same color per haplotype from (A). **(C)** Histogram of HR length distribution across different haplotypes in 101 non-mutated alleles (S3 Table). Each haplotype is represented by its specific color.

the haplotype and HR length was in line and complementary to the data from the 27 informative subjects (S3 Table). We found that K, F, and Q haplotypes harbored 2 HR units, whereas N and P contained 4 and 5 HR units, respectively. The J haplotype predominantly had 6 HR units (7 out of 9 tested alleles) and $R_h$ most frequently harbored 8 units (9 out of 26 tested alleles). A variable number of repeats were found in the other tested $R_h$ alleles (Fig 1C and S3

Table). Because of HR length variation in J and R haplotypes, we hereafter specify HR length for these haplotypes (*e.g.*, R haplotype with 12 units is designated $R_{12}$).

Altogether, we established a molecular map of the haplotypes and their HR length in Caucasians.

## Association between HR length and *C9orf72* gene expression

Reduced levels of normal *C9orf72* transcripts and proteins from the mutated allele can contribute to neural dysfunction in C9-ALS/FTD disease [27–29]. Hence, the expression level from the complementary non-mutated allele may be important for the overall C9orf72 protein level per cell and thus modify the disease phenotype. We therefore used our genetic map to search for possible associations between HR length and *C9orf72* expression from normal alleles.

It should be noted that a previous analysis of postmortem cerebellum samples from corticobasal degeneration patients revealed a positive correlation between V3 expression and the length of the longest HR allele, in a range of 2 to 28 repeats [30]. Yet this analysis measured the combined contribution of the two alleles and not the relative contribution of each allele to V3 expression.

To determine the relative allelic contribution to RNA transcripts, we used our SNP haplotype mapping for peak height analysis of heterozygous SNP sites in Sanger sequencing chromatograms of RT-PCR products. For accurate analysis, the peak height ratio of the two alternative nucleotides was corrected according to the ratio in which the two alleles contribute equally [31–33]. We verified the reliability and the validity of the method to analyze the relative contribution of two alleles (S3 Fig).

We analyzed skin fibroblasts from 12 individuals carrying K and/or P alleles who are therefore heterozygous for SNP rs10757668 in exon 2 and/or rs17769294 in exon 5 (Fig 2A and S2 Table). The second allele in these heterozygous individuals included all haplotypes except Q, and HR lengths of 2, 4, 5, 6, 7, 8, 10, 12, 16, and ~30 repeats. By using a forward primer in exon 1a (nt 59–80) that is common to both V1 and V3 and a reverse primer in exon 2 or 5 (Fig 2A, blue arrows, and S1 Table, #27, 28), we determined the relative allelic expression of these upstream promoter transcripts. Remarkably, in all fibroblast cultures, the allele with the longer HR expressed more combined V1 and V3 than its complementary allele with the shorter HR. Moreover, a linear correlation was observed between the HR-length ratio (*e.g.*, in a cell with 10 and 2 repeats, the ratio is 5) and the expression ratio between the two alleles (Fig 2B, $R^2 = 0.91$, $p = 1.8e\text{-}06$). We further evaluated the relation between HR length and V1 and V3 expression in $R_hF$ fibroblasts, by amplifying V1 and V3 transcripts that also carry an extended exon 5 with rs774359 G/A heterozygous site (Fig 2A, orange arrows, and S1 Table, #29). $R_h$ alleles (that contained 8–12 repeats) expression was dramatically higher than the complementary F alleles (2 repeats), supporting a positive effect of HR length on V1 and V3 expression (n = 4). We next evaluated the relation between HR length and V2 expression in fibroblasts by using the same methodology and heterozygous SNP sites (Fig 2A and S1 Table, #30–32). The HR length did not correlate significantly with V2 expression (Fig 2C, $R^2 = 0.22$, $p = 0.07$).

To evaluate if the increased HR length is associated with an increase in overall allelic *C9orf72* expression, we compared *C9orf72* expression between $R_h$ alleles with 7–16 repeats and their complementary K, F and P alleles, which carry 2–5 repeats. We observed a significant 1.25-fold increase in the overall allelic $R_h$ levels compared to the non-R alleles (Fig 2D, $p = 0.00016$).

Next, we determined the relation between large HR expansions and *C9orf72* expression in C9-ALS fibroblasts ($R_dK$ and $R_dF$; n = 5 and n = 3, respectively). For V1 and V3, $R_d$ allele expression was significantly higher than the K and F alleles in 7 out of 8 samples (Fig 2E), but

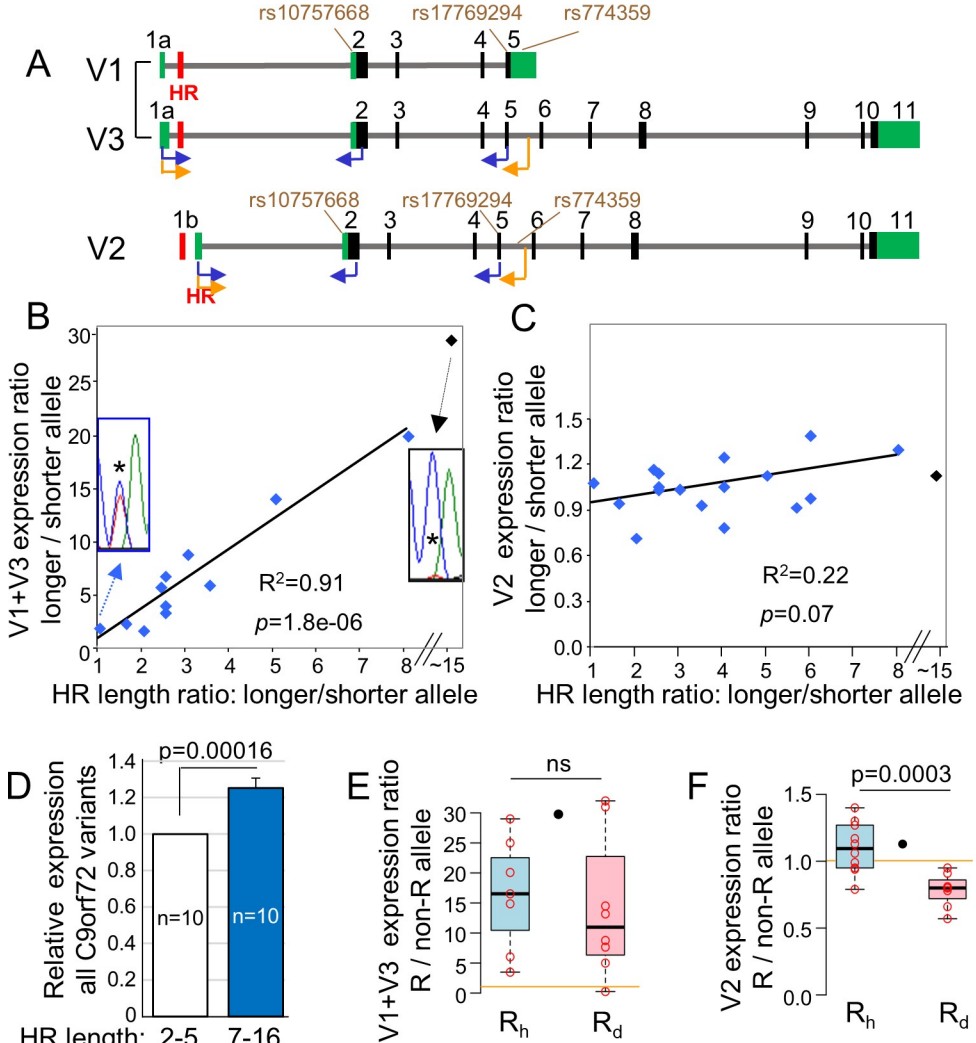

**Fig 2. Association between HR length and *C9orf72* gene expression. (A)** Schematic illustration of *C9orf72* variants analysis. Black and green blocks represent translated and untranslated exonal regions, respectively. RT-PCR primers (arrows) and SNPs used for analyses are indicated. **(B)** Scatterplot showing significant positive correlation between V1 +V3 expression level and HR length in normal and intermediate fibroblasts. The *x*-axis represents the HR length ratio between the longer to the shorter allele. The *y*-axis represents the expression ratio between the longer and the shorter allele. The line shows the best-fit regression line for the normal alleles. The intermediate $R_{30}K_2$ was not included in the equation due to its uncertain HR length. The inserts show the C/T heterozygous SNP rs10757668 (asterisk) within the RT-PCR amplified products of $F_2K_2$ (HR ratio of 1) and $R_{30}K_2$ (HR ratio of ~15). Blue peaks of the C nucleotide represent the F and the $R_{30}$ alleles, and red peaks of the T nucleotide represent the K allele. **(C)** Scatterplot showing non-significant correlation between V2 expression level and HR length in non-mutated fibroblasts. The *x*- and *y*-axes are as in (B). **(D)** The overall allelic *C9orf72* expression from $R_h$ alleles that carry 7–16 repeats compared to complementary F, K and P alleles, which carry 2–5 repeats. The allelic ratio determined by the analysis of heterozygous SNPs along exons 2, 5 and 11 (S1 Table, #12, 24 and 38). Data presented as mean ± SEM. **(E, F)** Boxplots showing the relative expression from $R_h$ allele with 7–16 repeat units (blue box) and from $R_d$ allele with expanded HR (pink box), relative to the complementary shorter alleles. The $R_{30}$ allele is depicted in black circle between the $R_h$ and the $R_d$ boxes. **(E)** V1+V3, **(F)** V2. Boxplots show median and quartiles and whiskers are 1.5 times the interquartile range. The orange horizontal line delineates a ratio of 1 and represents the non-R allele expression level. ns: non-significant.

did not significantly differ from the $R_h$ allele (Fig 2E). For V2, $R_d$ allele expression was significantly lower than the complementary K or F allele (Fig 2F; $R_d$ values lower than 1, $p = 0.0001$). Moreover, the ratio between $R_d$ alleles with large expansions and their complementary alleles

was significantly lower than between the $R_h$ alleles with 7–16 repeats and their complementary allele (Fig 2F, $p$ = 0.0003). Notably, for the intermediate $R_{30}$ allele, V2 expression was higher than its complementary K allele (Fig 2C and 2F, black dot). Thus, the effect of intermediate length of 30 repeats on *C9orf72* expression differs from large expansions, as previously reported [14].

## Association between HR length and introns 1 and 2 levels

There is uncertainty about the effect of *C9orf72* HR expansion on intron 1 levels [12,34–36]. We therefore performed RNA-seq to compare the RNA levels of *C9orf72* introns in control and C9-ALS cells. We previously generated iPSC lines from control and C9-ALS fibroblasts [37], which express *C9orf72* transcripts about three times more than skin fibroblasts (S4 Table), and are therefore more suitable for RNA-seq analysis (see **Materials and Methods** section for the methodology of analysis of relative RNA levels of introns).

In C9-ALS iPSCs, the mean normalized values for introns 1 and 2 were 3.1 and 2.7 times higher than control iPSCs, respectively (S4 Table and Fig 3A, $p$ = 6e-06 and 0.0002,

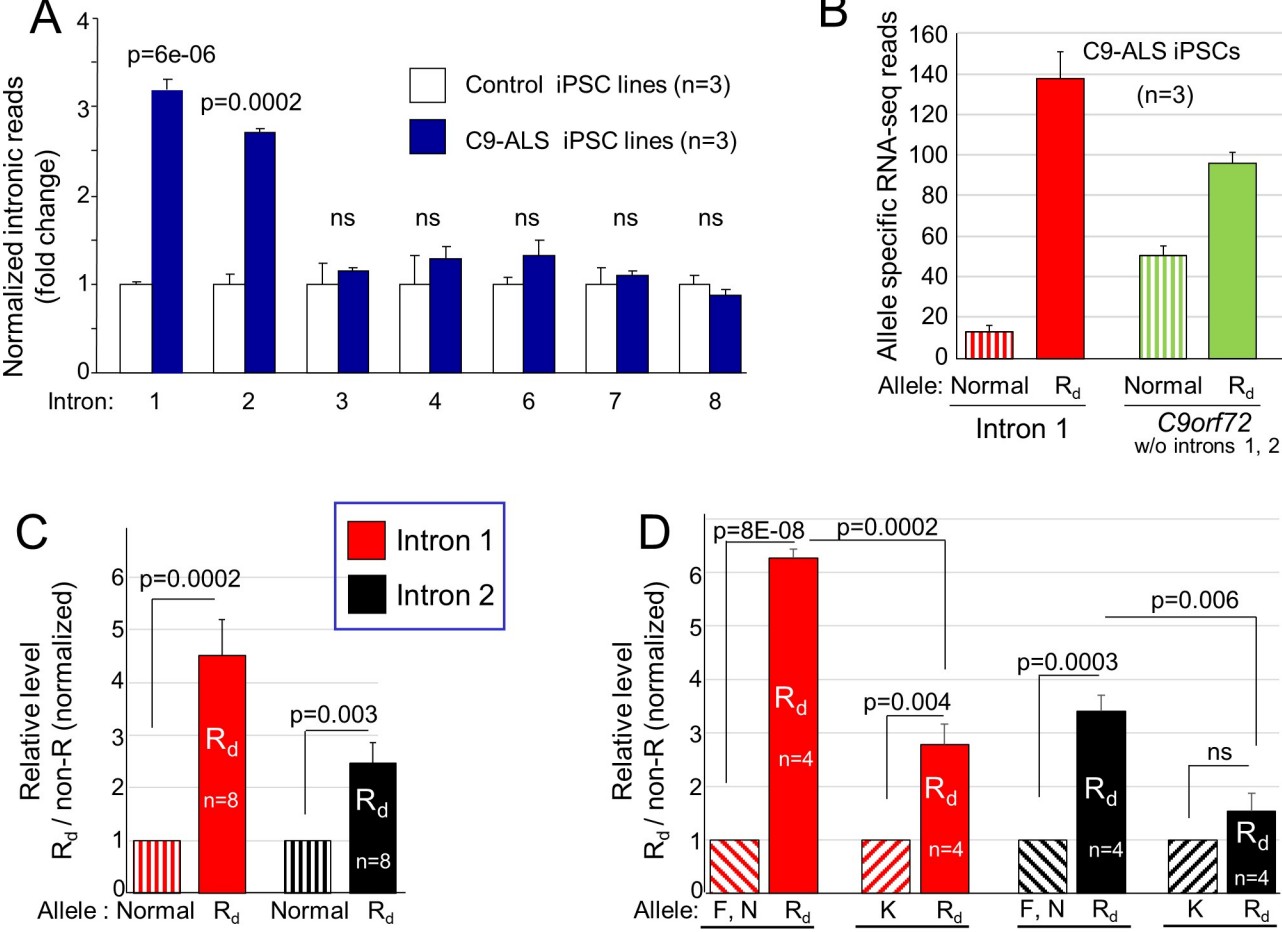

**Fig 3. Association between HR length and introns 1 and 2 levels. (A)** The relative frequency of RNA-seq reads aligned to *C9orf72* introns in control (white) and C9-ALS (blue) iPSC lines. Intron reads from each iPSC line were normalized to those of exons 2–5. Data represented as fold-change versus control iPSCs. **(B)** RNA-seq reads of C9-ALS iPSCs that overlap with heterozygous SNP sites are identifiable from their allelic origin from either the normal (dashed columns) or $R_d$ (solid columns) allele. Identifiable reads within intron 1 are depicted in red, and within other *C9orf72* regions (without intron 2) in green. **(C)** Allelic ratio in introns 1 and 2 of C9-ALS fibroblasts determined by Sanger sequencing of RT-PCR amplicons, which was normalized to the ratio in *C9orf72* exons. **(D)** The results in (C) displayed separately for the $R_d$K fibroblasts (n = 4) and for other C9-ALS fibroblasts ($R_d$F and $R_d$N; n = 4). Data presented as mean ± SEM. ns: non-significant.

respectively), while no significant accumulation was observed in other introns (Fig 3A). Similar results were obtained in RNA-seq analysis of skin fibroblasts (S4 Table).

Our genetic map allows us to identify the allelic origin of reads that overlap heterozygous SNP sites (S4 Fig). Intron 1 carries multiple SNP sites and is therefore appropriate for such an analysis. In C9-ALS iPSCs, a mean of 138 versus 13 identifiable intron 1 reads were derived from the $R_d$ and the normal allele, respectively, while for the rest of the gene, a mean of 96 reads derived from the $R_d$ allele compared with 51 from the normal allele (Fig 3B; n = 3). These results show that in intron 1 of our C9-ALS iPSCs, there is over a 10-fold increase in the number of $R_d$ reads versus its complementary allele. They also suggest that 1.9-fold of this increase is attributable to higher $R_d$ expression and 5.6-fold is probably attributable to inefficient $R_d$ splicing relative to the complementary allele.

We next analyzed the relative contribution of the $R_d$ allele to intron 1 in C9-ALS fibroblasts by using Sanger sequencing chromatograms of heterozygous SNP sites (see **Materials and Methods**). The $R_d$ allele contribution was 4.5-fold higher for intron 1 and 2.5-fold higher for intron 2 (Fig 3C; n = 8) but did not significantly differ in other introns that we examined (introns 3, 4, and 6). Since the analyses included amplicons that spanned intron 1-exon 2 and intron 2-exon 3 boundaries (S1 Table, #11 and 14), the relative increase in the $R_d$ allele might be attributed to a decrease in its splicing efficiency rather than to an accumulation of excised introns or lariat intermediates.

We next examined the results according to the identity of the complementary allele. We found that for $R_dK$ cells, the ratio between the $R_d$ to K allele was 2.8-fold for intron 1 ($p = 0.004$) and non-significant 1.5-fold higher for intron 2. In cells that did not carry the K allele ($R_dF$ and $R_dN$ cells), the $R_d$ allele was 6.2-fold higher for intron 1 ($p = 8e-08$) and 3.4-fold higher for intron 2 ($p = 0.0003$). Thus, the $R_d$ to K ratio was significantly lower than the $R_d$ to F and N ratios (Fig 3D, $p = 0.0002$ for intron 1, $p = 0.006$ for intron 2) indicating that intron 1 and 2 levels in K allele transcripts were higher than in other normal alleles.

To verify this observation, we analyzed normal fibroblast cultures that are heterozygous to the K allele. Remarkably, the K allele contribution to introns 1 and 2 was on average two times higher than the complementary allele (S5A Fig). The increased contribution of K allele was not observed for other introns that we analyzed (S6 Fig).

Interestingly, intron 1 levels of K allele were 2.6 times higher than complementary alleles with 2–7 repeats, but only 1.4 times higher than alleles with 10–30 repeats indicating higher intron 1 levels in moderate and intermediate repeat lengths (S5A Fig). Moreover, analysis of fibroblasts from multiple donors with RK haplotypes supported a correlation between the HR length and intron 1 levels (S7 Fig). To verify the effect of moderate HR lengths, we further analyzed introns 1 and 2 in $R_{12}F$ and $R_{12}P$ fibroblast cultures. Intron 1 RNA levels were 1.6-fold higher in the $R_{12}$ versus complementary $F_2$ and $P_5$ alleles (S5B Fig). Altogether, these results demonstrate that increased HR length is associated with higher intron 1 levels in normal and pathological ranges.

## HR expansion activates cryptic donor splice sites within intron 1

It was previously shown that *C9orf72* transcripts can use alternative donor splice sites in intron 1, downstream to exon 1b [35]. We therefore searched for alternative donor splice sites along intron 1 in both control and C9-ALS fibroblasts, by performing RT-PCR with several alternative forward primers along intron 1 and a reverse primer in exon 2 (S1 Table, #33–37). We identified four alternative splice donor sites in fibroblasts located 665, 910, 2295, and 2308 bp downstream to the V2 donor splice site at chr9: 27572766, 27572520, 27571136, and 27571123 (GRCh38/hg38), respectively (Fig 4A). These RT-PCR products were easily detected in

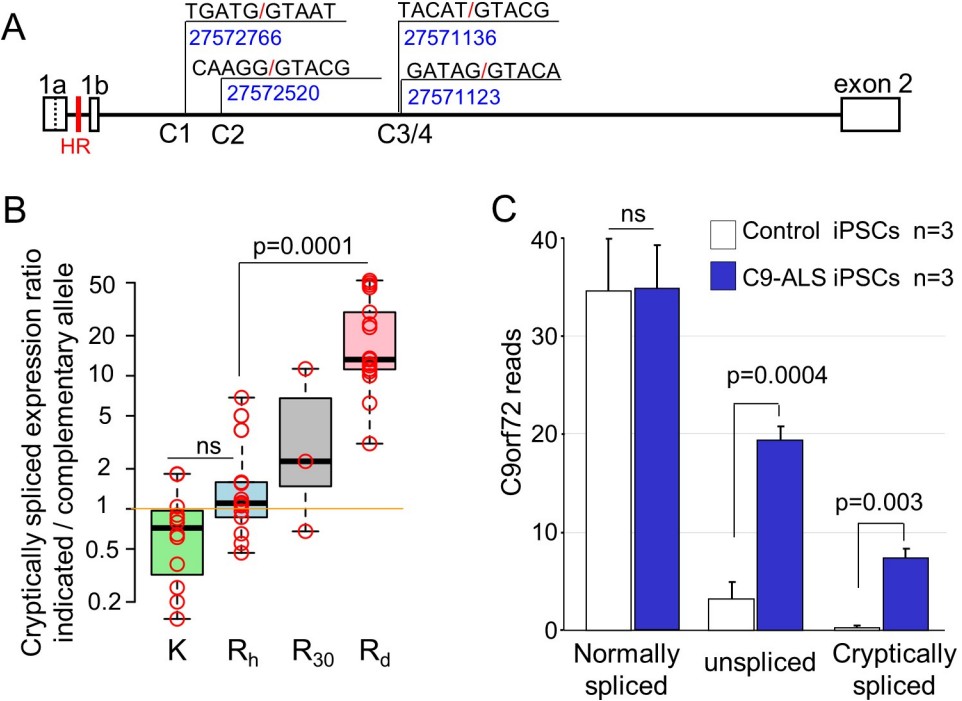

**Fig 4. HR expansion activates cryptic donor splice sites in intron 1. (A)** Schematic presentation for the location and sequence of intron 1 cryptic donor splice sites C1-C4. **(B)** Boxplots showing relative allelic usage of cryptic splice-sites (C1-C4) in fibroblasts. K alleles (relative to F, N, P, J, and $R_h$ alleles) are in green, $R_h$ alleles with 7–16 repeats (relative to K, F, and P alleles) are in blue, the $R_{30}$ allele (relative to K allele) is in grey, and the $R_d$ alleles (relative to K, F, and N alleles) are in pink. The orange horizontal line delineates a ratio of 1 and represents the expression level of the complementary allele. **(C)** The expression of normal, unspliced, and cryptically spliced *C9orf72* transcripts in control and C9-ALS iPSCs, based on the RNA-seq results. Data are presented as mean ± SEM. ns: non-significant.

C9-ALS cells but barely detectable in control cells. We therefore defined these splice sites as cryptic sites and named them C1, C2, C3, and C4, respectively (Fig 4A). C1 and C2 were analyzed separately, while C3 and C4 were analyzed together due to their proximity (13 bp) from each other. The relative allelic usage of these cryptic splice sites was determined according to the peak height ratio in heterozygous SNP sites (S8 Fig, asterisk).

In control fibroblasts, if cryptically spliced products were detectable, we frequently did not observe substantial differences between the two alleles (S8A and S8B Fig). Yet, a comparison between fibroblasts that carry heterozygous K and/or $R_h$ allele revealed a possible trend for higher cryptic transcript level in the $R_h$ versus K allele (Fig 4B, green and blue boxes). A similar non-conclusive trend was observed for D-478, who carried ~30 HR repeats (Fig 4B, grey box). Thus, it is possible that moderate and intermediate repeat lengths exert a minor effect on C1-C4 levels. In contrast, in C9-ALS fibroblasts that carry large HR expansions, most of C1-C4 transcripts were derived from the $R_d$ allele (Figs 4B(pink box) and S8C-E). Moreover, the ratio between R allele and its complementary non-R allele was significantly higher for the $R_d$ allele than for the $R_h$ allele (Fig 4B, *p* = 0.0001).

To improve the RNA-seq mapping algorithm's ability to identify reads originating from cryptic splicing events, we added C1-C4 to the Ensembl gene annotation file. In the three control iPSCs, we identified a single read containing a cryptic splice junction. In contrast, in the three C9-ALS iPSCs, we identified 24 reads containing a cryptic splice junction, and all but one joined to the 5'-end of exon 2 (Figs 4C and S9). Notably, eight of these reads that contained heterozygous SNP sites were all identified as $R_d$ allele products.

To evaluate the portion of unspliced and cryptically spliced transcripts from total *C9orf72* transcripts in normal and C9-ALS iPSCs, we counted reads that overlapped the 5' boundary of exon 2 and contained either the terminus of authentic or cryptic exons as well as unspliced intron transcripts (S9 Fig). Based on this analysis, we estimate that in the control non-mutated iPSCs, about 91% of *C9orf72* transcripts are normal while 8% are unspliced, and less than 1% are cryptically spliced. In contrast, in the C9-ALS iPSCs, only 57% of *C9orf72* transcripts are normal, whereas 31% and 12% are unspliced and cryptically spliced, respectively (Fig 4C).

## Discussion

To allow biallelic analysis of *C9orf72* gene transcripts, we generated a detailed genetic variation map of the *C9orf72* locus. We fully described the R haplotype and six other haplotypes (F, K, P, N, J and Q) that are common in Caucasians. In addition, we described the Z haplotype (Tables 1 and S2), which is common in China (>20%) and in India (>10%) but not among Europeans (~0.5%). Interestingly, among Asian C9-ALS patients, published genotypic analysis showed that neither allele matched the R haplotype. Thus, it was deduced that the diseased allele among Asian C9-ALS patients originated from a founder other than the one bearing the R haplotype [38–40]. However, these studies did not characterize the diseased Asian allele or identified its unique SNPs. Based on the limited published genotyping results of Asian patients [38–40], combined with our haplotype molecular map, it appears that the Z haplotype may be the Asian risk haplotype. However, to confirm this, additional studies are needed.

Our molecular mapping included the analysis of HR length in all haplotypes except haplotype Z. The HR length was constant in haplotypes harboring up to 5 repeat units, but not in haplotype J, which typically harbors 6 repeats. The highest level of HR length diversity was observed in haplotype R, which most frequently harbors 8 repeats (Fig 1C). These results are aligned with the reported correlation between tandem repeat length and the rate of change in length in subsequent generations [41,42]. Notably, the shortest documented HR length that changed in the subsequent generation was observed in a father and his daughter who had 11 and 12 repeats, respectively [6], whereas there are tremendous differences in pathological HR length between patient's cells and tissues [6,43–45].

We used our genetic map as a platform for biallelic expression analysis of cells from a single donor, instead of a more conventional comparison between cells from different donors with distinct genetic and environmental backgrounds, possibly with differing RNA or cDNA quality. Since the analysis is direct between the two alleles, it easily identifies unique allelic expression features, which are not mitigated but rather emphasized by the second conventional allele.

By analyzing the peak height ratio of two alternative nucleotides, we determined the relative allelic expression of *C9orf72* transcripts in fibroblasts. We showed that the expression of the upstream promoter transcripts, V1 and V3, increases with HR length in the range of 2-~30 repeats (Fig 2B). Our results are in accordance with RT-qPCR results observed in the cerebellum of corticobasal degeneration patients [30]. The levels of V1 and V3 expression from complementary non-mutated alleles may be relevant to the disease pathogenesis. Elevated V1 and V3 expression from non-mutated alleles is associated with an increase in overall allelic *C9orf72* expression (Fig 2D). Since a reduction in overall *C9orf72* expression from the $R_d$ allele can contribute to the C9-ALS/FTD disease [27–29], a complementary non-mutated allele with increased *C9orf72* expression might be beneficial to patients, when the $R_d$ allele is strongly repressed. On the other hand, the increased upstream promoter activity of the non-mutated allele, which is correlated with higher HR length within the normal range, is associated with

production of pre-spliced RNAs with a moderate HR length. This may aggravate the effects of the pathologically expanded HR RNAs.

Our study on $R_d$ allele transcription revealed some common features among C9-ALS patients. $R_d$ allele expression of V1 and V3 was higher than the complementary non-R allele in most samples, yet its expression levels was not higher than $R_h$ allele. V2 expression from $R_d$ alleles that carry a large but not intermediate expansion, was significantly lower than the complementary alleles, including the $R_h$ allele, consistent with previous studies [14,21,43]. Given the reduced V2 levels from $R_d$ compared to $R_h$ alleles, the similar V1 and V3 levels, combined with the high levels of cryptic and unspliced intron 1 transcripts from the $R_d$ allele (Fig 4C), it appears, as previously noted [13,21], that HR expansion in C9-ALS patients is associated with a marked shift in $R_d$ allele promoter utilization from the downstream harmless promoter to the upstream and potentially harmful promoter.

We further evaluated the effects of pathological HR expansion on intron 1 splicing. Since our research strategy was based on comparative biallelic analysis, we studied intron 1 splicing in normal haplotypes. We observed retention of intron 1 in the K haplotype, which potentially could result from SNP rs10757668G>A in exon 2, just 18 bp downstream of intron 1 (S10 Fig) [46,47]. Therefore, we stratified normal alleles into K and non-K alleles for accurate evaluation of intron 1-retention levels of the $R_d$ allele, when performing allelic evaluation (Fig 3D).

We showed in C9-ALS patients that there are higher unspliced intron 1 levels and activation of cryptic donor splice sites. These changes might be due to a repressive effect of the HR RNA on splicing factors at adjacent donor splice sites, resulting in HR containing transcripts that are either unspliced or cryptically spliced at sites that are more distal. Alternatively, these changes may reflect an adverse impact of the HR chromatin on the adjacent *C9orf72* V2 promoter that resulted in activation of more downstream transcription start sites that produce either unspliced or cryptically spliced HR-less transcripts. Previous experiments showed that antisense oligonucleotides (ASO) designed to bind downstream of the HR efficiently reduce the number of HR-containing nuclear foci [21,48,49], supporting the first explanation.

We further demonstrated that in both $R_d$ and K alleles, the increase in intron 1 levels is associated with an increase in intron 2 levels. This finding is not surprising, since mutation in a splicing site of one intron may affect the nearby intron [50]. Based on the number of RNA-seq reads in C9-ALS iPSCs (S4 Table) and the length of introns 1 and 2, we estimate that about 40% of the $R_d$ allele transcripts that contain an unspliced intron 1 also carry an unspliced intron 2.

Notably, our results regarding *C9orf72* transcripts and cryptic splicing were observed in fibroblasts and iPSCs, and it will be important to evaluate the generalizability of the results in MNs and, more preferably, MNs derived from iPSCs of ALS patients.

ASO [21,48,49,51] and CRISPR manipulations [52] have great therapeutic potential for C9-ALS/FTD. However, such therapies may also affect *C9orf72* transcripts that originate from the normal allele. Reducing C9orf72 protein levels increases inflammatory conditions [53,54] and increases toxicity of dipeptide repeat proteins [20,28]. ASO designed against the HR sequence may confer a preferential targeting against the $R_d$ allele [51]. However, since GGGGCC repeats are found in multiple genes [55], an ASO that targets the pathogenic repeat may reduce the expression of multiple genes. Therefore, ASO exploiting heterozygous SNP sites for specifically targeting the $R_d$ allele and minimize interference to normal allele transcripts might be the best strategy, as previously demonstrated in Huntington disease [56–58]. Based on our results, introns 1 and 2 might be good candidates for allelic specific targeting, while the 5' region of intron 1, which is also retained in cryptically spliced transcripts, might be the best target region. Notably, since the repertoire of the heterozygous SNP sites in patients

depends on the identity of the complementary allele, the preferred target site might differ between patients.

Our present molecular map of the *C9orf72* gene (Tables 1 and S2) spans a relatively short genomic region of about 30 kb that exhibits a relatively low recombination rate of ~0.35 cM/Mb (centiMorgans per megabase). Since recombination events along this region are rare, most of the Caucasian alleles are of "pure" haplotype (S3 Table) and the allelic origin of heterozygous SNPs can be easily determined. Our map is straightforward, user friendly, and enables the study of specific effects of $R_d$ and other alleles, design of allelic-specific ASOs, and evaluation of the impact of drugs and ASOs on allelic expression of beneficial transcripts such as V2, and harmful transcripts such as unspliced and cryptic transcripts.

In a wider perspective, deciphering the major haplotypes for other genes, and the formation of a straightforward and accessible molecular map, may improve a researcher's ability to determine effects of genetic variations and to identify imprinted genes. Most importantly, this may enable specific targeting of toxic alleles in protein misfolding disorders.

In summary, we generated a detailed genetic map of the major *C9orf72* haplotypes and applied polymorphic sites to determine the effects of HR length and SNP variation on *C9orf72* transcript expression and splicing in normal and C9-ALS cells. The characterization of *C9orf72* genetic variations may pave the way to allele-specific therapeutic strategies.

## Materials and methods

### Ethics statement

The ALS patients and the healthy control participants signed an informed consent that included genetic analysis of acquired biospecimens. The study was approved by the University of Michigan Medical School Institutional Review Board (Protocol # HUM00028826). Established lines HEK293, HES-1 (a human embryonic stem cell line; NIH code ES01), OL (a kind gift from Michel Revel) and FOR002 (both are foreskin lines) were used to correlate the HR length with specific haplotypes.

### Fibroblast isolation

Human skin fibroblast cultures were established from punch biopsies from Caucasian control and ALS patients from the University of Michigan ALS Clinic. Biopsies were placed in fibroblast growth media (GM) [high-glucose DMEM supplemented with 10% fetal calf serum, 1× Glutamax-1 and 1× MEM NEAA (Gibco/Thermo Fisher Scientific, Waltham, MA)]. Tissues were washed three times and then cut into small pieces that were resuspended in 0.5 ml GM and plated in a minimal amount of GM in two T25 flasks in a humidified atmosphere with 5% $CO_2$ at 37˚C. At day 3 and day 6, one ml of GM was added to each flask. Keratinocytes migrated out of the tissue pieces at day 7, and fibroblasts migrated out at day 10 or later. At ~day 21, the cells in both T25 flasks were trypsinized with 0.25% trypsin-EDTA (Gibco) and split 1:5 into 100 mm dishes (Falcon or Corning). After 7–10 days, the fibroblasts reached ~85% confluency and were trypsinized to make 9 working stock vials in freezing medium [GM + 10% DMSO (Sigma-Aldrich, St Louis, MO)], which were stored in liquid nitrogen in collaboration with the Michigan ALS Consortium at the University of Michigan.

### Cell culture

Skin fibroblasts were cultured in high-glucose DMEM (Invitrogen/Thermo Fisher Scientific) supplemented with 15% fetal calf serum (Biological Industries, Beit Haemek, Israel), 2 mM L-glutamine, 50 U/ml penicillin, and 50 μg/ml streptomycin (all from Invitrogen). iPSC lines

were maintained on mitomycin-C (MMC, 10 μg/ml; Sigma-Aldrich)-treated human foreskin fibroblasts in gelatin-coated 6-well plates ($3 \times 10^5$ feeders/well; Nunc, Roskilde, Denmark) cultured in hESC medium, which consisted of Knockout DMEM supplemented with 16% Knock-Out SR, 2 mM L-glutamine, 1% nonessential amino acids, 0.1 mM β-mercaptoethanol, 50 U/ml penicillin, 50 μg/ml streptomycin (all from Invitrogen), and 5 ng/ml bFGF (PeproTech, Rocky Hill, NJ) in a 5% $O_2$ incubator. iPSCs were passaged weekly by mechanical dissection or dissociation with 1 mg/ml collagenase IV (Gibco).

## Genomic DNA extraction

Skin fibroblasts were washed with PBS and dissociated with 0.04% trypsin–0.17 mM EDTA (Invitrogen). Following centrifugation, cells were resuspended to an estimated cell concentration of $10^6$ cells/ml in PBS. Five microliters of the resuspended cells were added to 15 μl pre-chilled lysis buffer (final concentration: 10 mM Tris-HCl, pH 8.3, 50 mM KCl, 0.1% Triton X-100, 0.1% Tween 20) with 0.2 μl proteinase K (Roche) in 0.2 ml Eppendorf tube. Cells were then incubated at 56˚C for 2 h followed by 96˚C for 25 min and 15˚C for 15 min. For iPSC analysis, half of an iPSC colony was washed with PBS, collected in 5 μl PBS, and treated as described for fibroblasts.

## RNA isolation and cDNA preparation

Total RNA was isolated from fibroblasts (passages 4–7) and iPSCs (passages 6–10) using a PerfectPure RNA Cultured Cell Kit (5Prime, Gaithersburg, MD) following the manufacturer's protocol, including on-column DNase treatment. For RT-PCR, 2 μg of total RNA was reverse transcribed using the High-Capacity cDNA RT kit (Applied Biosystems/Thermo Fisher Scientific) with the provided random hexamer primers. Reaction mixtures were prepared with and without RT to confirm there was no detectable genomic DNA contamination in the samples.

## PCR analysis

PCR amplifications were performed with 2× PCRBIO HS Taq Mix (PCR Biosystems, United Kingdom), KAPA HiFi HotStart ReadyMix (2×, KAPA Biosystems/Roche), and Phusion Hot Start High-Fidelity DNA polymerase (Finnzymes/Thermo Fisher Scientific), using the primers listed in S1 Table. PCR products were excised from agarose gels and extracted using Wizard SV Gel and PCR Clean-Up System (Promega, Madison, WI). Amplicons were Sanger sequenced using BigDye Terminator v1.1 Cycle Sequencing Kit and 3730xl DNA Analyzer (Applied Biosystems, CA). Sequence Scanner Software 2 (Applied Biosystems, CA) was used for sequence data analyses. To calculate the allelic expression ratio, the peak height ratio of the two alternative nucleotides within a cDNA amplified product was normalized to the peak height ratio measured in genomic DNA PCR samples, which represents a 1:1 allelic ratio. At the sites with allele-specific background, the allelic background noise was subtracted from the measured peak height, while samples with high background noise were not included in the analyses. All observations were validated by alternative SNP sites and primer sets. To evaluate the retention levels of introns, the expression ratios in heterozygous SNPs along introns were normalized to the ratio in heterozygous SNPs along exons 2, 5, and 11 (S1 Table, #12, 24, and 38). To avoid misinterpreting the RT-PCR results due to co-localization of intron sense sequences and antisense transcripts derived from the opposite strand [35], intron 1 analyses was performed at least 0.5 kb downstream to the HR (S1 Table, #4–11), where the level of antisense transcripts is very low (S11 Fig).

Fluorescent fragment length analysis of PCR amplified fragments containing the HR was performed as previously described ([59]; laboratory B method) using the KAPA2G Robust

PCR Kit and buffer A (KAPA Biosystems). The sequences of the corresponding primers are listed in S1 Table, #3.

## RNA sequencing

RNA integrity was confirmed on the 2200 TapeStation (Agilent Technologies, Santa Clara, CA) and quantified with Qubit RNA HS Assay Kit (Invitrogen). Only RNA with a RNA integrity number (RIN) $\geq$ 9.4 was used for RNA-seq. To retain immature transcripts and introns in the analysis, the poly(A) mRNA enrichment step was omitted. Instead, rRNA was depleted from 1 μg of total RNA using the Ribo-Zero rRNA Removal Kit (Epicentre Biotechnologies/ Illumina, Madison, WI). Barcoded cDNA libraries were constructed using the RNA Library Prep Kit v2 (Illumina, San Diego, CA) or the KAPA Stranded mRNA-Seq Kit (KAPA Biosystems) according to the manufacturer protocol and sequenced using NextSeq 500 System, High-Output mode 75 cycles kit (Illumina).

Raw reads were quality-trimmed at both ends with a quality threshold of 32, then adapter sequences were removed with Cutadapt (version 1.9.1; https://cutadapt.readthedocs.io/en/v1.9.1/index.html), only retaining reads at least 15 nt long. Low quality reads were then filtered using the FASTQ Quality Filter program of the FASTX-toolkit (version 0.0.13; http://hannonlab.cshl.edu/fastx_toolkit/download.html), with parameters–q20 –p90. Processed reads were mapped to the GRCh38 human reference genome using TopHat (v2.0.14; https://ccb.jhu.edu/software/tophat/index.shtml). Mapping used gene annotations from Ensembl (release 78), together with the C1-C4 splice sites information to identify reads that originated from these genomic locations as *C9orf72* splicing variants. The relative abundance of abnormal transcripts to total *C9orf72* transcripts was determined. Reads that overlap the 5' boundary of exon 2 were counted and identified as an authentic variant, a cryptically spliced transcript, or an unspliced transcript. To overcome bias against the *C9orf72* 5' end [21], the average coverage per base along exons 2–5 was taken as an exon 2 boundary, and the portion of authentic transcripts (V1, V2, and V3 altogether) from the total transcripts was evaluated by subtracting the unspliced and cryptically spliced transcripts from the average value. To evaluate the retention levels of *C9orf72* introns, intronic read counts were normalized with the read counts along exons 2–5, which are shared by all *C9orf72* transcripts (S1 Fig). The RNA-seq data has been deposited at the GEO database, accession number GSE162943 (http://www.ncbi.nlm.nih.gov/geo).

## Statistical analysis

Data are expressed as the mean ± SEM. Statistical comparisons of means were performed by a two-tailed unpaired Student's *t*-test. The value of $p \leq 0.05$ was considered significant. Boxplots were produced using BoxPlotR web-tool (http://shiny.chemgrid.org/boxplotr/) and represent median values (black line), interquartile ranges (colored regions), and Tukey whiskers (define values within 1.5 times the interquartile range from the upper and lower quartile). *P*-values corresponding to the regression coefficients were computed using MATLAB (MathWorks, Natick, MA).

## Supporting information

**S1 Fig. Schematic illustration of *C9orf72* variants V1, V2, and V3.** *C9orf72* variants named as in PubMed and the UCSC Genome. Black and green blocks represent translated and untranslated exonic regions, respectively, and red triangles represent the HR site. (PDF)

**S2 Fig. Fluorescence fragment length analyses of PCR fragments containing the HR.** The length of each amplified product is indicated. The number of HR units was calculated by subtracting the length of the 5' and 3' flanking regions (total 118 bp) from the PCR product length and dividing by six. **(A)** PCR amplified products of control H-1021 (homozygous for $R_h$ alleles) are 178 and 166 bp long, indicating HR repeats of 10 and 8 units ($R_{10}R_8$). **(B)** PCR amplified product of the $R_dK$-ALS patient D-540. The 130 bp fragment reveals HR length of 2 units at the K allele. The length of the $R_d$ allele in D-540 was previously determined by Southern blot analysis as 350–390 repeats and is not PCR amplified. **(C)** PCR amplified product from $R_dK$ ALS patient D-478. The 130 bp fragment reveals HR length of 2 units in the K allele. The R allele HR length appears as sawtooth peaks of 20–35 repeats and a maximal peak of 27 repeats.
(PDF)

**S3 Fig. Sanger sequencing can determine the relative expression of two alleles. (A)** Representative Sanger sequencing chromatogram of equimolar amounts of two 659 bp DNA fragments that differ in a single nucleotide (C versus T in rs10757668; S1 Table, #11). The peak height of T is higher than C. The PCR fragments originated from H-1021 and H-1116 fibroblasts that carry RR and KK haplotypes, respectively (S3 Table). **(B)** Representative Sanger sequencing chromatograms of different mixtures of two 659 bp long DNA fragments (S1 Table, #11) that differ in a single nucleotide (C versus T). **(C)** Mixture series of opposite homozygous DNA fragments as in (B) demonstrate a linear relationship between the relative allelic contribution and the relative peak height. Peak-height ratios (C/C+T) were normalized according to the 1:1 allelic ratio shown in (A) and plotted against the actual concentration ratios. The plot shows 35 data points obtained from three or four independent sequencing runs for each mixture. Some points are co-localized and appear as a single point. The points are fitted with best-fit linear regression line and shown together with the $R^2$ value and the calculated equation for the slope. **(D)** T/C peaks height ratio in equimolar mixture of opposite homozygous DNA fragments as in (A) compared to PCR amplification of heterozygous genomic DNA. The peak height ratio in the two groups is similar. ns: non-significant.
(PDF)

**S4 Fig. RNA-seq reads that overlap with heterozygous SNP sites are identifiable for their allelic origin.** Representative RNA-seq reads from C9-ALS iPSCs (D-312). The reads spanning the heterozygous position of SNP rs3849945 in the first intron of *C9orf72* gene. Thirteen of the identifiable reads originated from the $R_d$ allele and carry G nucleotide (orange), while only one carry the reference A nucleotide (indicated by green arrow), which originated from the complementary F allele.
(PDF)

**S5 Fig. K allele and increased normal HR length are associated with increased intron 1 and 2 levels. (A)** Allelic analysis of introns 1 and 2 in normal and intermediate fibroblasts that carry K allele and a complementary allele with 2-~30 repeats (n = 9). The ratio in the intronic SNPs was normalized to the ratios obtained in exons 2, 5, and 11 (S1 Table, #12, 24 and 38). The dashed columns represent the expression levels of the K alleles and the left solid columns represent the relative expression of the complementary alleles. Analysis according to the length of the non-K allele reveals higher accumulation of intron 1 in lengths of 10-~30 versus 2–7 repeats. **(B)** Allelic analysis of introns 1 and 2 in normal $R_{12}F$ and $R_{12}P$ fibroblasts. The ratio in intronic SNPs was normalized to the ratios obtained in exons 2, 5, and 11 (S1 Table, #12, 24 and 38). The contribution of $R_{12}$ allele to introns 1 is higher than the complementary F and P alleles that carry 2 and 5 repeats, respectively. Data presented as mean ± SEM. ns: non-

significant.
(PDF)

**S6 Fig. K allele is associated with increased introns 1 and 2 levels.** A representative set of Sanger sequencing chromatograms for different regions of the *C9orf72* gene. All chromatograms represent amplification products from control H-966 fibroblasts that carry the K and P alleles (S3 Table). Each chromatogram in the figure includes five sequential nucleotides with a central polymorphic nucleotide. The alternative nucleotide color of K and P alleles is indicated by a colored letter above the chromatograms. For each chromatogram pair, the left and the right panels represent amplification products of genomic DNA and cDNA, respectively. All SNPs shown in the figure are described in S2 Table. The PCR primers used, and amplicon lengths are described in S1 Table. The relative level of the K allele is markedly higher in introns 1 and 2 but not in other introns, nor in exons. The K allele also contributed more to the amplified products that spanned intron 1-exon 2 and exon 2-intron 3 boundaries. Therefore, the relative increase in the K allele probably attributed to a reduction in its splicing efficiency.
(PDF)

**S7 Fig. HR length is associated with introns 1 and 2 levels in RK cells.** Fibroblasts that carry both R and K alleles but differ in the HR length of the R allele were analyzed for R/K allelic ratio in introns 1 (red) and intron 2 (black). The orange horizontal line delineates a ratio of 1 and represents the K allele intronic levels.
(PDF)

**S8 Fig. Determination of the allelic contribution to *C9orf72* cryptic splice products by Sanger sequencing.** Fibroblast of control donors **(A, B)** and C9-ALS patients **(C-E)** were RT-PCR amplified for cryptic splice site products (C1-C4) and Sanger sequenced with a reverse primer in exon 2. The representative chromatograms in the figure show the complementary strand of the products described in Fig 4A. All cells carry a single K haplotype and therefore are heterozygous for rs10757668 (C/T), marked by an asterisk. In normal cells, the relative contribution of two normal alleles is frequently similar (A, B); while in C9-ALS cells, the vast majority of C1-C4 transcripts are the $R_d$ allele products. C3 and C4 were co-amplified in (E).
(PDF)

**S9 Fig. Identification and quantification of unspliced and cryptically spliced transcripts by using RNA-seq data. (A)** RNA-seq mapping results for intron 1-exon 2 boundary region. The frequency of reads spanning intron 1-exon 2 junction (unspliced) is higher in C9-ALS than in control iPSCs. **(B)** RNA-seq mapping results in a representative C9-ALS iPSC line. Reads overlapping cryptic splice junction are indicated (C1-C4). Black arrows indicate the boundary between intron 1 and exon 2.
(PDF)

**S10 Fig. Possible mechanisms of intron 1 retention in the K haplotype.** Substitution of G to A in SNP rs10757668 (TTTGGATAA to TTTAGATAA) is specific to K haplotype. This SNP is in exon 2, just 18 bp downstream of intron 1. **(A)** It may account for destroying a putative TGGA exonic splicing enhancer site [47]. **(B)** Alternatively, it may account for the formation of exonic splicing silencer site resembling the hnRNP A1 binding site (TTTAGACAA) in the *SMN2* gene [46].
(PDF)

**S11 Fig. Distribution of sense and antisense reads along the 5' region of *C9orf72*.** Examples for strand-specific RNA-seq analysis of control and C9-ALS cells. Reads of sense and antisense

(AS) transcripts in the 5' region of the gene *C9orf72* gene (~8 kb) are illustrated in pink and blue, respectively. The green line indicates the HR site. Most transcripts that are located between exon 1b and exon 2 are sense transcripts.
(PDF)

**S1 Table. PCR primer pairs used for SNP analysis in this study.**
(DOCX)

**S2 Table. Detailed genetic map of the *C9orf72* locus.**
(DOCX)

**S3 Table. Haplotype and HR length results of the research participants.**
(DOCX)

**S4 Table. RNA-seq reads of the *C9orf72* gene in iPSCs and fibroblasts from control and C9-ALS patients.**
(DOCX)

**S1 Data. Numerical data that underlines graphs.**
(XLSX)

## Acknowledgments

We thank the late Alfred Taubman and The Taubman Foundation for their kind donation. We also thank Judy and Sidney Swartz and the Swartz foundation for their generous and continues support of all our activities. We thank The Sinai Medical Staff Foundation for their support. We thank Masha G. Savelieff, Michal Gropp, and Shelly Tannenbaum for editing the manuscript, Shai Carmi for helpful discussions, Shilo Rosenwasser for insights into statistics analysis, Idit Shiff and Sharona Elgavish for help with RNA-seq, and Idit Amir and Aviad Zick for providing genomic deep sequencing data.

## Author Contributions

**Conceptualization:** Israel Ben-Dor, Benjamin E. Reubinoff.

**Data curation:** Israel Ben-Dor, Yuval Nevo.

**Formal analysis:** Israel Ben-Dor, Yuval Nevo.

**Funding acquisition:** Eva L. Feldman, Benjamin E. Reubinoff.

**Investigation:** Israel Ben-Dor.

**Methodology:** Israel Ben-Dor, Eva L. Feldman.

**Project administration:** Israel Ben-Dor, Eva L. Feldman, Benjamin E. Reubinoff.

**Resources:** Crystal Pacut, Eva L. Feldman, Benjamin E. Reubinoff.

**Software:** Yuval Nevo.

**Supervision:** Eva L. Feldman, Benjamin E. Reubinoff.

**Validation:** Israel Ben-Dor, Benjamin E. Reubinoff.

**Visualization:** Israel Ben-Dor.

**Writing – original draft:** Israel Ben-Dor, Benjamin E. Reubinoff.

**Writing – review & editing:** Israel Ben-Dor, Eva L. Feldman, Benjamin E. Reubinoff.

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
