## [Decision Letter · Decision Letter 0]

9 Sep 2020

Dear Dr Reubinoff,

Thank you very much for submitting your Research Article entitled 'Characterization of C9orf72 haplotypes to evaluate the effects of normal and pathological variations on its expression and splicing' to PLOS Genetics. Your manuscript was fully evaluated at the editorial level and by independent peer reviewers. The reviewers appreciated the attention to an important problem, but raised some substantial concerns about the current manuscript. Based on the reviews, we will not be able to accept this version of the manuscript, but we would be willing to review again a much-revised version. We cannot, of course, promise publication at that time.

If you decide to revise the manuscript for further consideration at PLOS Genetics, please aim to resubmit within the next 60 days, unless it will take extra time to address the concerns of the reviewers, in which case we would appreciate an expected resubmission date by email to plosgenetics@plos.org.

[LINK]

We are sorry that we cannot be more positive about your manuscript at this stage. Please do not hesitate to contact us if you have any concerns or questions.

Yours sincerely,

Giorgio Sirugo

Associate Editor

PLOS Genetics

Hua Tang

Section Editor: Natural Variation

PLOS Genetics

Reviewer's Responses to Questions

**Comments to the Authors:**

Reviewer #1: The authors examined the allele around C9ORF72 in detail and produced a detailed map of its site. Based on their results, they examined its gene expression and showed that a haplotype allele with a long HR affects the expression of that gene and splicing of intron 1.

Descriptive in content, I find no real problem with the experimental methods or the results. The significance of the results of this study is discussed, as well as the implications of the results for future ASO therapies and so on, which I believe is certainly a possibility.

The biggest problem is the lack of focus on the purpose of this study. The main objective is not conveyed to the reader.

The search for pathologically increased HR-specific haplotypes would be significant, but the analysis of pathologically increased haplotypes was done in 12 ALS patients, not a sufficient number of analyses for haplotype analysis. In the end, they ended up analyzing normal haplotypes within the normal range and are related to variations in the number of repeats. It is certainly worthwhile to understand the actual situation, but if that is the purpose of the study, I think it would be better to focus the content on it.

The part that examines the effect of this augmented allele on transcription is also abrupt because of the flow of this paper; I think it is necessary to show how much of the V1 and V3 allele is expressed to the V2 allele, and then discuss whether or not it affects the pathology. As currently described, it is unclear what the significance of this change is.

Thus, in conclusion, this result may be useful in the ASO setting, but it is unclear what significance the effect on transcription will have. The first half of the paper and the second half of the paper even seem to be dissociated.

It could be said that this paper has analyzed the haplotypes of the normal allele in detail and used the results to examine the effect of the allele on the expression of the same gene in a sample of ALS patients who were heterozygous. However, the paper is poorly structured and does not adequately convey its significance to the reader. I think it would be better to significantly change the structure of the paper and focus on a clear objective.

Reviewer #2: I Ben-Dor and coll. have performed an interesting study on C9orf72 haplotypes and on the expression of C9orf72 variants (RNA) as a function of haplotypes and the presence or absence of pathological hexanucleotide expansions. These data are important for future fundamental studies and possibly for the development of therapeutics, but I have several criticisms:

Major comments :

-The authors did not indicate whether they obtained a signed consent for the analysis of their genetic characteristics. Furthermore, it would be interesting to have more information about these Americans of European descent: are the authors certain that they can consider them as "Europeans" especially if they have been Americans for several generations? If this is not possible please discuss this in the results and discussion sections. On page 8, the authors indicate that they have identified a "Z" haplotype, if I understand this discovery was made by another team or study. If this is the case please cite the reference or origin. I don't think this is somewhere in the manuscript.

-Could the author indicate if the patients studied on page 11 (called "European subjects": 15 and 30 subjects) are the same as those used for the description of haplotypes.

-the authors used (RT-)PCR/Sanger sequencing for gene expression studies. Please indicate on page 12 that these results were obtained on the transcripts and therefore are studies on RT-PCR products. The authors have purified their RT-PCR products to set up the technique and to produce Figure 2B. Was the purification step always performed for the study of patients and healthy individuals? Why did the authors not use allele-specific RT-PCR or RNAseq for this work? I would like to know if this relative quantification technique by RT-PCR/Sanger is reproducible: for example, are the peak heights (ratios) the same if the experiment is repeated three times on the same RNA sample?

-I propose that the authors reduce the size of the section "HR length and haplotype effects on introns 1 and 2 levels".

-page 17: is the "heterozygous SNP site" cited in the second paragraph of page 7 present on transcripts V1, V2 and V3, or only V1 and V3?

-Did the authors investigate whether the use of a cryptic donor splice site resulted in a change of the acceptor splice site?

-it would be important that the authors indicate in the discussion section that the observations they made should be replicated in neurons, or (better) motor neurons such as motor neurons derived from iPSCs of patients or healthy individuals. Indeed the expression levels of the transcription variants could be very different compared to fibroblasts or iPSCs, isnt’it? Same for cryptic sites.

Minor comments :

- Page 12: We aimed to determine the possible effects of haplotype and HR length on C9orf72. Perhaps “relations” could be better than “effects”, as this work does not allow us to observe the mechanistic effects.

- Fibroblast cultures: could the authors indicate the passage number of the cultures. The age of the cultures has an impact on the functioning (gene expression) of the cells.

- I propose to place table 2 in the supplementary data (it is fully described in the text).

-Figure 1: replace “Our Sample” by “American European population (our sample)”

Reviewer #3: This manuscript reports a rather thorough study of C9orf72 haplotypes in the transcribed region, in relation to length of the hexanucleotide repeat (HR), and the effect of length of repeat on expression at the RNA level in fibroblast, in iPSCs of ALS patients or controls (n=3 for each). The authors investigated alternate promoter use, intron retention and activation of cryptic donor splice sites distal to HR in intron one. In all of these, they could use the haplotype specific SNPs in the transcript they have identified in their haplotype analysis to distinguish expression from each allele.

However, I thought the paper quite lengthy, and the importance of the observations for the pathogenic mechanisms in ALS rather unclear or confirmatory (the founder effect) in some cases. For instance the effect of number of repeats on V1+V3 expression is very striking in the normal range (Fig3B) but what may be the relation to ALS mechanisms, as the Rd alleles have actually lower (but non significantly) relative expression than the Rh (7-16 repeats (Fig3D).

One positive aspect is certainly that the identification of transcribed SNPs specific for the R haplotype will allow to study better in an allelic specific way the cis effect of expansion in patient samples.

As I am not a specialist of c9orf studies, I will ask a question that may be valid or not, but I think it would be interesting to have the answer of the authors. All the abundant c9orf literature concentrate on the effect of HR expansion on expression of the c9orf72 gene at RNA or protein level (DNA methylation, toxic RNA, toxic dipeptides..). However, there are 2 nearby genes INFK and MOB3B whose promoters are at about 50 or 44kb from the HR. There are several instances of intronic SNPs that actually affect the expression of the nearby gene for instance FTO/IRX3, Smemo et al Nature 2014, Lactase/MCM6 Labrie et al Nature Struct Mol Bio 2016. Were there any studies testing the potential effect of HR expansion on expression of these two genes. As the at-risk haplotype extends in most cases to the transcribed regions of INFK and MOB3B, would it be possible to look in the same allelic specific way to the effect of HR length in normal or expanded alleles?. Would the RNAseq experiments carried here allow to test this hypothesis?

A second also naïve questions: regarding the founder effect: there are different possible scenarios. The longer repeat length around 8 may be by itself sufficient to generate in a highly recurrent way larger alleles that themselves become more highly unstable, and one would thus not expect any major sub-haplotypes in patients. However, an alternate possibility is that of a single or rather small number of events that generated a longer repeat, for instance 12 to 15, and these secondary alleles are really the precursors: in this case one could expect to have a few rarer SNVs (too rare to be studied by standard SNP array strategy) in patients, associated to the haplotype carrying the pathogenic expansions. Was this investigated? Would the authors data allow to test this hypothesis.

The authors state in abstract, author summary, and twice in discussion (p25, where there is a bit of redundancy), including the last sentence, that the R haplotype characterization may be invaluable or pave the way for allele specific therapeutic strategies. However, given the huge difference in length of the pathogenic repeat versus non pathogenic, would a repeat based targeting strategy be actually simpler (or would it suffer from difficulties linked to the GC only sequence?)

Fig4 Did the authors investigate whether intron retention is modified when the iPSCs are differentiated to neuronal precursors or neurons?

Editorial suggestions

Table 1 takes 2 pages and details all the tested SNPs in respect to the 8 haplotypes. However, for the vast majority of researchers working on c9orf because of its link with ALS; the table in the main text could be reduced to those SNPs specific for the R haplotype and the rest could be in a supplementary table. Also, I counted in the table 6 R specific SNPs, while there seems to be 10 or 11 in Fig 1B: what is the reason for this discrepancy?

Finally, it took me some effort to try to correlate to the R haplotype specific SNPs described previously by Smith BN et al (2013) or Goldstein O et al Neurobiol Aging 2018, not cited). It seems that most or all of these previous SNPs fall outside the region investigated here and thus of the transcribed region? I suggest that either in the shortened table 1 or in Fig1a the authors position these previous R specific SNPs on the map

Fig2 is a control of validity for a rather standard methodology, and it could be in supp info , as most of page 12

The effect of K haplotype is described in 1 page in results, and half a page in discussion. While the data are convincing, is there any hint that this could be relevant to ALS risk? Are there studies in ALS that might point to an effect on risk either in trans to an expanded repeat, or as a risk factor for sporadic ALS without expanded allele. Otherwise, the length given to this perhaps anecdotic observation seems too large.

**Have all data underlying the figures and results presented in the manuscript been provided?**

Reviewer #1: Yes

Reviewer #2: Yes

Reviewer #3: None

PLOS authors have the option to publish the peer review history of their article (what does this mean?). If published, this will include your full peer review and any attached files.

Reviewer #1: No

Reviewer #2: No

Reviewer #3: No

---

## [Decision Letter · Decision Letter 1]

25 Feb 2021

Dear Dr Reubinoff,

We are pleased to inform you that your manuscript entitled "Characterization of C9orf72 haplotypes to evaluate the effects of normal and pathological variations on its expression and splicing" has been editorially accepted for publication in PLOS Genetics. Congratulations!

Yours sincerely,

Giorgio Sirugo

Associate Editor

PLOS Genetics

Hua Tang

Section Editor: Natural Variation

PLOS Genetics

Comments from the reviewers (if applicable):

Reviewer's Responses to Questions

**Comments to the Authors:**

Reviewer #1: They responded to my request adequately.

Reviewer #2: no comment

**Have all data underlying the figures and results presented in the manuscript been provided?**

Reviewer #1: Yes

Reviewer #2: Yes

PLOS authors have the option to publish the peer review history of their article (what does this mean?). If published, this will include your full peer review and any attached files.

Reviewer #1: **Yes: **Osamu Onodera MD, PhD.

Reviewer #2: No

**Data Deposition**

http://datadryad.org/submit?journalID=pgenetics&manu=PGENETICS-D-20-01137R1

**Press Queries**

---

## [Editor Report · Acceptance letter]

23 Mar 2021

PGENETICS-D-20-01137R1 

Characterization of C9orf72 haplotypes to evaluate the effects of normal and pathological variations on its expression and splicing 

Dear Dr Reubinoff, 

We are pleased to inform you that your manuscript entitled "Characterization of C9orf72 haplotypes to evaluate the effects of normal and pathological variations on its expression and splicing" has been formally accepted for publication in PLOS Genetics! Your manuscript is now with our production department and you will be notified of the publication date in due course.

With kind regards,

Alice Ellingham

PLOS Genetics

On behalf of:
